# Management of Enterovesical Fistula in a Patient with Crohn’s Disease: A Case Report and Literature Review

**DOI:** 10.3390/diagnostics13091527

**Published:** 2023-04-24

**Authors:** Ming-Wei Hsu, Wen-Chi Chen, Ting-Na Wei, Chi-Ping Huang

**Affiliations:** 1Department of Urology, China Medical University Hospital, Taichung 404327, Taiwan; d29950@mail.cmuh.org.tw (M.-W.H.); d0283@mail.cmuh.org.tw (W.-C.C.); 2Graduate Institute of Integrated Medicine, College of Chinese Medicine, China Medical University, Taichung 404328, Taiwan; 3Department of Radiation Oncology, Taichung Veterans General Hospital, Taichung 40705, Taiwan; 4School of Medicine, China Medical University, Taichung 404328, Taiwan

**Keywords:** Crohn’s disease, enterovesical fistula, urinary tract infection, surgery, computed tomography

## Abstract

Enterovesical fistula (EVF) is a rare complication of Crohn’s disease (CD), characterized by recurrent urinary tract infections, fecaluria, and pneumaturia. However, most diagnostic tools have low sensitivity for EVF. Management consists of conservative and surgical approaches. Conservative treatment is usually adopted first. However, the appropriate time to consider surgery remains controversial. Herein, we report on the case of a 34-year-old male who presented with diffuse abdominal pain with fullness for one day. Enteroscopy and biopsy confirmed the diagnosis of Crohn’s disease. Contrast-enhanced computed tomography (CT) suggested a fistula between the ileum and urinary bladder; however, cystoscopy did not find an obvious tract. The patient initially received medical treatment, but the symptoms persisted with recurrent urinary tract infections and subsequent bilateral hydronephrosis. He then underwent successful fistulectomy, partial cystectomy, and two segmental resections of the small bowel with end-to-end primary sutures. No complications or symptomatic urinary tract infections were noted during 30 months of follow-up after surgery, suggesting no recurrence of EVF. Surgical intervention is warranted when medical treatment fails or complications occur. Clinical symptoms and laboratory data are often less informative for the diagnosis of EVF, and CT is the most helpful diagnostic modality. Our management strategy provides an option for such patients.

## 1. Introduction

Crohn’s disease (CD) is a chronic inflammatory disease of the gastrointestinal tract, and intestinal complications such as fistulae may occur as a result of the overactivated immune response. Perianal, enterovesical, colovesical, enterovaginal, rectovaginal, and enterocutaneous fistula have been reported in about one-third of patients with CD [1]. The incidence of enterovesical fistula (EVF) is rare, with a reported incidence of 2–5% among patients with CD. EVF refers to a passageway connecting the bowel and urinary bladder, and its distinguishing features from other fistulas include recurrent urinary tract infections, fecaluria, and pneumaturia [2]. The diagnosis is usually difficult, and abdominopelvic computed tomography (CT) and cystoscopy are valuable tools for both diagnosis and surgical planning [3].

There are two choices of management for EVF: conservative and surgical approaches. Patients are often initially treated medically, especially those with poor physical status and advanced cancer [3]. Although surgical intervention seems to have a better prognosis based on limited studies, stiff and frail bladder tissue often present a challenge in surgery [4,5]. Therefore, the timing of surgical treatment remains unclear. Herein, we report on a challenging case of a patient with CD complicated with EVF who was successfully treated with surgery after the failure of medical therapy. We also discuss surgical management and diagnostic tools.

## 2. Case Report

A 34-year-old man with a medical history of anal fistula was admitted to the emergency department because of diffuse abdominal pain with fullness for one day. A physical examination revealed a distended abdomen with hypoactive bowel sounds. There was mild rebound tenderness without muscle guarding. His body temperature was 37.6 °C, and blood pressure and other vital signs were within the normal range. A blood test showed leukocytosis with a white blood cell count of 18,600/μL (normal range 3.6–11.2 × 10^3^/μL) and serum C-reactive protein level of 13.34 mg/dL (normal range <0.8 mg/dL). Urine analysis disclosed a red blood cell count of 349/μL (normal range <17/μL) and white blood cell count of >1000/μL (normal range <28/μL). A urine culture yielded *E. coli* and *K. pneumoniae* with extended-spectrum β-lactamases. Abdominopelvic CT (Figure 1) implied the high likelihood of the presence of inflammatory bowel disease and a potential fistula between the ileum and urinary bladder. Enteroscopy showed skipped deep and longitudinal ulcers with bowel stricture over the proximal and terminal ileum. Skipped deep ulcers with some polyps were also noted from the rectum to the ascending colon. CD was verified from a histopathologic report.

Due to highly suspected EVF, cystoscopy and cystography were performed. Cystoscopy (Figure 2) revealed some edematous and irregular mucosa over the posterior wall but no obvious fistula. A floppy guidewire failed to find a fistulous tract, and no contrast leakage was detected on cystography. In addition, a barium enema showed no strong imaging evidence of EVF.

The patient was initially treated with conservative treatment with the administration of the anti-inflammatory agent mesalamine 1000 mg three times daily and prednisolone 10 mg three times daily. However, he experienced persistent abdominal pain and recurrent urinary tract infection one month later. We then added azathioprine at a daily dose of 50 mg, along with an induction therapy of adalimumab administered subcutaneously at a dosage of 160 mg, followed by 80 mg after two weeks, and subsequently 40 mg every two weeks thereafter. Antibiotics were prescribed according to the urine culture. After adjusting the prescription, dysuria and frequent urination accompanied by pyuria in the urine analysis were still noted, indicating a recurrent urinary tract infection. In addition, the patient’s renal function deteriorated from an estimated glomerular filtration rate (eGFR) of 143 (creatinine: 0.64 mg/dL) to 71 (creatinine: 1.17 mg/dL) mL/min/1.73 m^2^ in one year. Abdominopelvic CT revealed bilateral hydronephrosis and hydroureter resulting from ureterovesical junction obstruction, and EVF could still not be excluded (Figure 3). Double-J stent placement was not advised due to the recurrent urinary tract infection, and the patient refused to receive percutaneous nephrostomy. Because of the highly suspected EVF, poor quality of life, and exacerbation of renal function, a surgical intervention with exploratory laparotomy was performed. Prior to the surgery, we gradually reduced the medication regimen to mesalazine at a dose of 1000 mg twice daily, azathioprine at 50 mg per day, and adalimumab at 40 mg every two weeks. Intraoperatively, an enlarged bladder with thickened mucosa was observed, and an EVF tract was found located on the posterior bladder wall. The fistula was resected, followed by partial cystectomy. The bladder was closed with a single-layer interrupted suture, and we left an indwelling urinary catheter. In addition, we performed two segmental resections and anastomoses of the small bowel (Figure 4). After the surgery, we resumed the use of biological agents along with mesalazine at a dosage of 500 mg twice daily and azathioprine at 50 mg per day. Adalimumab was later replaced with vedolizumab at a monthly dosage of 300 mg via intravenous infusion. His renal function then improved, with an increase in eGFR from 58 to 131 mL/min/1.73 m^2^ at 30 months of follow-up. Neither complications nor recurrent urinary tract infections were detected after surgery.

## 3. Discussion

A fistula is an abnormal tract that connects two different epithelial surfaces. In addition, epithelial-to-mesenchymal transition, an inflammatory process in which breakdown and repair of the intestinal wall with adjacent organs occurs due to the release of various cytokines and proteases, is thought to be associated with fistula formation in patients with CD [6]. In our patient, inflammation occurred through the entire thickness of the bowel wall, in which an abscess formed between the small intestine and urinary bladder. Over time, the abscess “eroded” the bladder and the fistula developed.

To date, the diagnosis of EVF is still challenging, and it is made harder by the lack of a gold standard for EVF evaluation. In addition, the poor diagnostic ability of known imaging studies makes identification of the fistula difficult. Thus, the diagnosis of EVF is primarily based on clinical evidence and supported by imaging findings. The most common symptoms, such as urinary tract infections and pneumaturia, are highly suggestive of EVF. Traditional diagnostic modalities include barium enema, colonoscopy, cystoscopy, and cystography.

Barium enema can help determine the potential cause of EVF, including malignancy and diverticulitis; however, the detection efficacy for fistula is limited due to a low sensitivity of 30–35% [7]. Lower gastrointestinal endoscopy is most helpful in diagnosing the underlying pathology of EVF, especially for malignancy. Likewise, it is challenging to visualize the presence of an EVF, with a detection rate of between 5 and 55% in published studies [8,9,10].

Cystoscopy is considered to have an increased diagnostic yield, and it has the ability to locate the possible fistulous tract by identifying its surrounding area characteristic of erythema, edema, and congestion. Furthermore, it can help exclude possible etiologies of EVF, such as urological malignancies, bladder stones, and interstitial cystitis [11]. However, it is usually hard to visualize the orifice and even the lumen of the fistula endoscopically unless the tract is very wide. Cystography may indicate the existence of a fistula by showing contrast outside the bladder. Likewise, the fistulous tract may be invisible as the edematous tissue can lead to closure of the fistula. In such cases, the EVF cannot be identified through barium enema, cystoscopy, or cystography. Taken together, fistula swelling may narrow the lumen, making it more difficult for contrast media to enter the lumen.

Compared to other diagnostic tools, CT is the most reliable technique owing to its high sensitivity (up to 90%) in the detection of EVF [12]. The most common findings are gas in the urinary bladder, thickened bladder wall adjacent to the edematous bowel wall, and adherence of the soft tissue mass between the urinary bladder and the intestine. More importantly, CT can suggest not only the etiology but also the anatomic structure around the fistula, which can help when planning an operation [3,11].

Magnetic resonance imaging (MRI) has been reported to be able to accurately depict the fistulous tract due to its excellent intrinsic soft tissue resolution, and it may be warranted in patients who are contraindicated for CT or in difficult cases such as complex fistulae. However, whether MRI is superior to CT remains controversial, and further large prospective studies are needed to elucidate this issue [13]. In our opinion, CT is still the modality of choice because of its cost effectiveness and availability. For patients suspected of having EVF, CT is necessary to provide more detailed information.

In patients with CD complicated with EVF, management is mainly divided into medical therapy and surgical approaches. Medical treatment consists of antibiotics; 5-ASA compounds such as mesalazine, azathioprine, and systemic corticosteroids; and anti-TNF therapy such as infliximab and adalimumab. Medical treatment has been reported to achieve long-term remission in up to 35% of patients based on three studies [14,15,16]. However, the rates of morbidity with poor physical status, progression of malignant disease, and septic complications from the EVF are higher after medical therapy compared to a surgical approach. Some authors have reported that conservative treatment should be reserved for those who are not suitable for surgery due to a poor condition, intolerance to general anesthesia, or terminal disease [17]. Nevertheless, several factors have been linked to an increased need for surgery. In a retrospective study of 37 CD patients with EVF, Zhang et al. reported that small bowel obstruction, abscess formation, enterocutaneous fistula, refractory urinary tract infection, and persistent ureteral obstruction were significant risk factors for surgery [14]. Therefore, medical treatment is insufficient in some circumstances, and further operative interventions should be considered.

Recurrent infections can cause fragility and thickness of the bladder and intestinal walls, which increases the operative time and complications. Nevertheless, surgical treatment can still achieve a higher remission rate in the long term. In the study reported by Yamamoto et al. [15], 25 of 30 EVFs in patients with CD were ultimately treated surgically. Among them, no sequelae or fistulae recurrence occurred in 22 of the patients after a median follow-up of 13 years. In addition, Taxonera et al. [16]. reported that 78 of 79 patients with CD-related EVFs who were treated with surgery had symptom relief during a mean follow-up of 101 months. Taken together, surgery seems to provide a better solution compared to medical treatment.

To compare the effectiveness of different diagnostic tools and treatments in patients with EVF, we searched PubMed 2.0 and Google Scholar using the keywords “Crohn’s disease” and “enterovesical fistula” and excluded articles without full text and those not in English. We identified 19 articles in the last 20 years after eliminating studies without complete information. The details of the articles, including the patients’ characteristics, diagnosis, and outcomes of conservative and surgical managements, are shown in Table 1. Over the past 20 years, patients with EVF have generally been between 30 and 50 years old, with the location of the fistula predominantly in the ileum, which is the most common site of occurrence for Crohn’s disease. Other locations have included the sigmoid, rectum, and even the jejunum. The most commonly used diagnostic tools to detect fistula in the reviewed cases were cystoscopy, CT scan, and MRI, which is compatible with our literature review. Medical treatments for EVF may involve antibiotics, immunosuppressants, corticosteroids, and biological agents. However, there is incomplete data regarding peri-operative medications in previous cases, and the prescriptions depended on the individual’s condition.

In most cases, patients underwent surgery, which included bowel resection and fistulectomy with bladder defect repair, using either laparotomy or laparoscopic approaches. Surgical intervention was found to achieve remission rates of more than 90%, while approximately 20% of patients were cured through conservative treatment. After reviewing the articles, surgery appears to be the preferable treatment option for eligible patients with EVF.

The basic concept of surgery is to excise the fistula, including the involved bladder wall and bowel segments, followed by repair. There are different approaches for this. Primary resection and anastomosis of the intestine during the same procedure without an enteric diversion is called the single-stage strategy, and it is currently the most employed procedure [32]. This approach is associated with low mortality and avoids unnecessary staged surgery. Stage surgery is often recommended in high-risk patients who cannot tolerate the surgery well [2,17]. Our surgical procedure was also performed in the same way.

With regard to bladder repair, partial cystectomy with a free margin is necessary in patients with suspected neoplasia to reduce the recurrence rate. However, there is no difference between excision or oversewing of the bladder with small defects as it can heal spontaneously [33]. An indwelling urinary catheter should be maintained for 7–15 postoperative days during the healing process. Our results confirmed this method.

Our patient underwent a successful exploratory laparotomy. The fistulae was close to the trigon of the bladder, and the inflammatory bladder wall was thick and stiff, making the surgery more difficult. Laparoscopic approach is also an option for EVF according to previous reports; however, laparotomy is a better choice in such a challenging case. At 30 months postsurgery, no recurrence was noted and his renal function improved.

Perioperative pharmacological considerations were also important in this patient. Prior to surgery, we gradually tapered down mesalazine and discontinued corticosteroids to minimize the risk of impaired wound healing, elevated infections, and bone marrow suppression [34]. After surgery, we resumed a reduced dose of immunosuppressant and biological agents to prevent the recurrence of EVF and continue treatment of CD. There are currently no guidelines on when to perform exploratory laparotomy, and previous studies have reported that surgical options are more likely to be considered after medical treatment fails. In our case, there were three indications for exploratory laparotomy. First, cystitis caused by recurrent urinary tract infections resulted in bladder wall thickening, which induced distal ureteral obstruction and hydronephrosis. In addition, lower urinary tract symptoms of frequent urination and dysuria progressed. These symptoms severely affected his daily life. Second, his renal function deteriorated from an eGFR of 143 (creatinine: 0.64 mg/dL) to 71 (creatinine: 1.17 mg/dL) mL/min/1.73 m^2^ within one year. Third, medical treatment failed and even caused aplastic anemia, a side effect of mesalazine. Our limited experience provides a treatment option for CD patients with EVF.

## 4. Conclusions

The diagnosis of EVF is challenging and is primarily based on clinical symptoms and supported by imaging findings. CT is one of the most important diagnostic tools, which can both localize the fistula and also assist in making a surgical plan. Surgical intervention is imperative in some situations, such as progressive lower urinary tract symptoms, deteriorating renal function, or poor response to medical treatment. The long-term remission rate of EVF appears to be higher with an operative approach than a medical therapy approach.

## Figures and Tables

**Figure 1 diagnostics-13-01527-f001:**
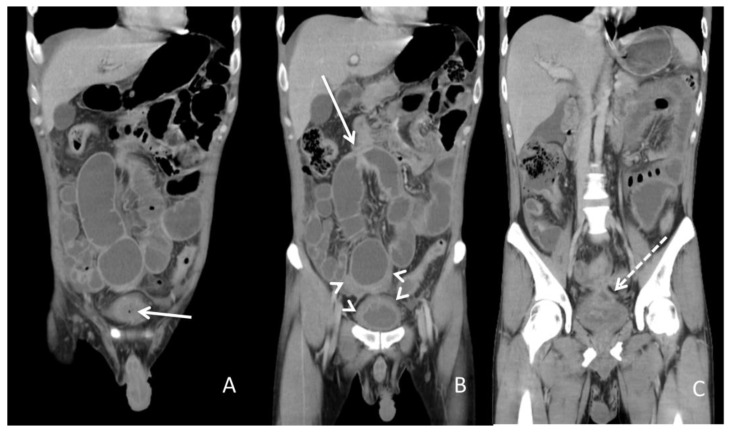
Abdominopelvic computed tomography. (**A**) The solid arrow shows a small air bubble in the urinary bladder. (**B**) The solid arrow shows one of the dilated small intestines with stricture measuring approximately 4 cm in length. The arrowheads show a thickened bladder wall adjacent to the inflammatory intestine. (**C**) The dotted arrow shows suspicious fistula between the small intestine and urinary bladder.

**Figure 2 diagnostics-13-01527-f002:**
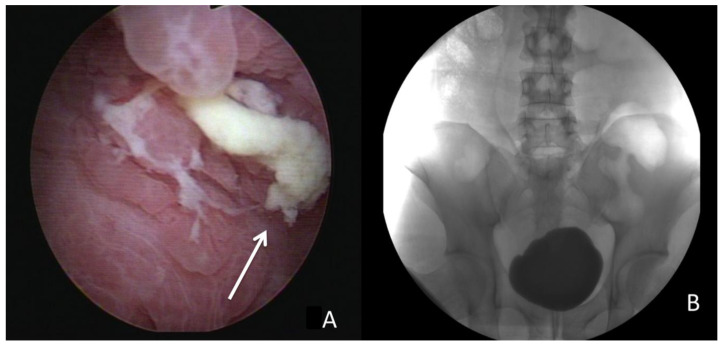
(**A**) Cystoscopy revealed some debris attached to the inflammatory mucosa (solid arrow). (**B**) Cystography showed no leakage after 200 mL contrast instillation.

**Figure 3 diagnostics-13-01527-f003:**
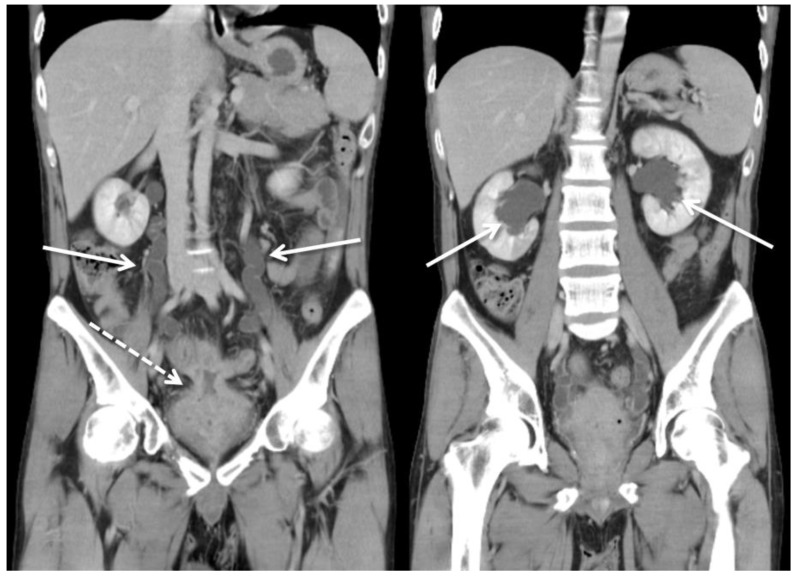
Computed tomography showed ureterovesical junction obstruction due to an increased amount of soft tissue (dotted arrow) in the pelvis, between the small intestine and urinary bladder. The solid arrow shows severe bilateral hydronephrosis and hydroureter.

**Figure 4 diagnostics-13-01527-f004:**
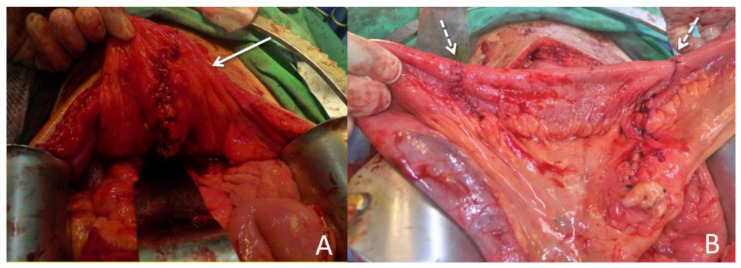
Surgical intervention. (**A**) The patient underwent a resection of the fistula, along with partial cystectomy. The solid arrow indicates the repaired urinary bladder. (**B**) Two segmental resections of the small bowel with end-to-end primary sutures (dotted arrow).

**Table 1 diagnostics-13-01527-t001:** Studies of enterovesical fistula in Crohn’s disease.

No.	Author	Year	Cases	Median Age (Years)/Sex *	Fistula Origin	Diagnostic Tool for Fistula	Outcomes of Conservative Treatment	Cases with Surgery	Surgical Outcomes
1	Yamamoto [15]	2000	30	39/ M:F = 2:1	Ileum (76.7%) Sigmoid (20%) Ileorectal anastomosis (3.3%)	Radiographically (*n* = 8) Cystoscopically (*n* = 16)	16.7% asymptomatic over 13 years; 83.3% converted to surgery	83.30%	10% had postoperative sepsis followed by recurrent EVF over 13 years
2	Miehsler [18]	2003	1	32/M	Ileum and sigmoid	Cystoscopy	Failed and converted to surgery	100%	No recurrence over 3 years
3	Chebli [19]	2004	1	23/F	Ileum	Barium enema	Failed and converted to surgery	100%	No recurrence over 6 months
4	Fukuda [20]	2005	1	35/F	Sigmoid	Barium enema	No recurrence 1 month later	No surgery	No data
5	Fischetti [21]	2007	4	Unknown	Unknown	Not mentioned	All failed and converted to surgery	100%	No recurrence
6	Ferguson [22]	2008	22	54.3/ Unknown	Ileum or sigmoid	Not mentioned	Not mentioned	100%	No recurrence over 26.4 months
7	Mizushima [23]	2012	1	51/F	Ileum	CT scan	Not mentioned	Laparoscopic surgery	No recurrence over 25 months
8	Cullis [24]	2013	1	13/M	Ileum	MR and barium	Took medications after surgery	100%	No recurrence over 1 year
9	Zhang [14]	2014	37	32/ M = 21; F = 16	Ileum (78.4%) Sigmoid and both (21.6%)	Not mentioned	35.1% remission over 4.7 years; 64.9% converted to surgery	64.9%	No recurrence over 3.9 years
10	Su [2]	2014	4	27.5/M	Ileum (75%) Sigmoid (25%)	MRI (*n* = 3) CT scan (*n* = 1) Cystoscopy (*n* = 4) Colonoscopy (*n* = 1)	50% remission over 3 years; 50% converted to surgery	50%	No recurrence over 20 months
11	Taxonera [16]	2016	97	33/ M:F = 3:1	Ileum (64.9%) Colon (23.7%) Rectum (7.2%) Jejunum (2%)	MRI (59.8%) CT scan (53.6%) Cystoscopy (10.3%) Surgery (8.2%)	17.5% remission over 91 months; 81.4% converted to surgery	Laparotomy (69%) Laparoscopic surgery (12.4%)	98.7% remission over 101 months
12	Vagianos [5]	2017	9	42/ M = 8; F = 1	Ileum (33.3%) Ileocecal (66.7%)	MRI plus cystography (*n* = 3)	All failed and 77.8% converted to surgery	77.8%	No recurrence over 42 months
13	Moniuszko [25]	2018	1	52/F	Unknown	Cystoscopy	Failed and converted to surgery	Stem cell transplantation	No recurrence over 36 months
14	Ye [26]	2019	1	79/M	Ileum	Cystoscopy and CT scan	Not mentioned	100%	Another fistula one year later
15	Nevo [27]	2019	7	Unknown	Ileum (100%)	CT scan	All failed and converted to surgery	Laparoscopic surgery	No recurrence over 49 months
16	de Groof [28]	2019	16	29.5/ M:F = 1:1	Ileum (81.2%) Colon (18.8%)	CT scan	Not mentioned	100%	No recurrence over 123.4 months
17	McKenna [29]	2022	44	46.5/ M:63%	Ileum (64%) Sigmoid (27%) Ileum and sigmoid (9%)	Not mentioned	50% took biochemical agents after surgery	Laparotomy (66%) Laparoscopic surgery (34%)	No recurrence over 17 months
18	Li [30]	2022	1	38/M	Rectum	Colonoscopy and MRI	No recurrence 3 months later	No surgery	No data
19	Gadiyaram [31]	2022	1	34/M	Ileum	CT scan	Took medications after surgery	Laparoscopic surgery	No recurrence over 2 years

* M = male; F = female.

## Data Availability

All of the data are available upon request to the corresponding author.

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
