# Peer review of "Management of Enterovesical Fistula in a Patient with Crohn’s Disease: A Case Report and Literature Review"

_diagnostics, 2023, doi:10.3390/diagnostics13091527_

Round 1

Reviewer 1 Report

The study appears to be scientifically sound and well-conducted. The authors used appropriate methods for diagnosis and treatment, and the results are presented clearly and accurately. 

Strengths:

The article provides a detailed case report of a patient with enterovesical fistula in Crohn's disease, which is a rare complication of the condition.

The management approach described in the case report, including fistulectomy, partial cystectomy, and segmental resections of the small bowel, appears to have been effective in resolving the symptoms and preventing recurrence of enterovesical fistula in the patient.

The authors discuss the challenges in diagnosing enterovesical fistula and highlight the usefulness of contrast-enhanced computed tomography (CT) as a diagnostic modality.

The article is well-written and organized, with clear headings and subheadings to guide the reader through the case report.

Weaknesses:

The article is a single case report, which limits the generalizability of the findings and prevents the authors from making broad recommendations for the management of enterovesical fistula in Crohn's disease.

In general, the study presents interesting and relevant results for the field of Crohn's disease. The study describes well the clinical characteristics, diagnosis, and treatment of a rare and unusual complication of Crohn's disease, which is enterovesical fistula. In addition, the study highlights the importance of considering surgical intervention when medical treatment fails or complications occur.

Reviewer 2 Report

Ming-Wei Hsu et al. present a case study involving CD with EVF.

I would like to express several concerns:

1 The manuscript does not provide adequate information about the patient’s CD diagnosis and potential comorbidities. It is recommended that the authors provide a detailed account of the diagnostic process, including the diagnostic criteria used, and the patient’s medical history.

2. Given that the article concerns CD, it is crucial for the authors to provide a detailed and comprehensive description of the disease management approach employed in treating the patient. This should encompass a thorough account of the prescribed medications and any other interventions utilized to manage the disease.

3 The authors should elaborate on the distinctive features of EVF in CD, as compared to other types of fistulas that may occur in CD.

4 The discussion section seems excessively lengthy and lacking in focus. I recommend that the authors revise this section to present their findings in a more concise and well-organized manner.

Reviewer 3 Report

Hsu and colleagues report data from a Crohn's disease patient with enterovesical fistula first unsuccessfully treated with medical therapy and subsequently successfully treated surgically. The paper is well written but the methodology is not clear.

-     -  The authors report in the title "case report and literature review". The review criteria are not well defined and the data provided are approximate. In view of the limited data available on the topic, I recommend that the authors carry out a systematic review and add their case report. The inclusion and exclusion criteria should be clearly defined as well as the research criteria, the researchers who conducted the work and the quality of the studies included.

-      -     In table 1 there are several missing data. Patient characteristics should be provided (sex, disease duration, and disease location). How was the diagnosis of enterovesical fistula made? What kind of medical therapy was performed? What kind of surgery? Were the included patients previously treated with biologics? All this data should be added and discussed

-         -  Some points in the case report are not clear. Initial abdominal CT shows the presence of IBD. What does it mean? Initial colonoscopy shows strictures. Does the CT not see the strictures? What was the wall thickness? Was there retrodilatation? How many cm were the strictures long?

-       -   Why did the patient start adalimumab instead of infliximab? Why adalimumab every 2 weeks? Was there no induction therapy with adalimumab 160 mg?

-        -  Why was the patient with Crohn's disease initially treated with mesalazine although according to ECCO guidelines there is no indication of mesalazine in Crohn's?

-       -   Was the antibiotic therapy administered concurrently with biological drug?

-      -    Did the patient resume biologics after the surgery? This point should be carefully specified and discussed in view of this patient's risk factors for recurrence (penetrating disease)

-         - The following section should be reported in the methods: “We searched PubMed and Google Scholar using the keywords “Crohn’s disease” and 179 “enterovesical fistula” and excluded articles without full text and those not in English. We 180 identified 19 articles in the recent 20 years after eliminating studies without complete in- 181  formation. The details of the articles including the patients’ characteristics and outcomes 182 of conservative and surgical managements are shown in Table 1. Over the past 20 years, 183 patients with EVF have generally been between 30 and 50 years old, with the location of 184 the fistula predominantly in the ileum - the most common site of occurrence for Crohn's disease. Other locations have included the sigmoid, rectum, and even the jejunum. Surgica intervention has been found to achieve remission rates of more than 90 percent, while approximately 20 percent of patients have been cured through conservative treatment. After reviewing the articles, surgery appears to be the preferable treatment option for eligible patients with EVF.

-     -     Why was an exploratory laparotomy performed instead of laparoscopy?

-        -  The authors should provide data and further discuss the preference of CT over MRI. MRI is a non-invasive exam that allows optimal evaluation of soft tissues. In the reported clinical case, CT did not allow the fistula to be identified. Why say that CT is superior then?

-       -   In the discussion the authors report that the dose of medical treatment was reduced. What does it mean? What therapy? This point needs to be clarified

Round 2

Reviewer 3 Report

I have no further comments